# PET Molecular Imaging: A Holistic Review of Current Practice and Emerging Perspectives for Diagnosis, Therapeutic Evaluation and Prognosis in Clinical Oncology

**DOI:** 10.3390/ijms22084159

**Published:** 2021-04-16

**Authors:** Valentin Duclos, Alex Iep, Léa Gomez, Lucas Goldfarb, Florent L. Besson

**Affiliations:** 1Department of Biophysics and Nuclear Medicine-Molecular Imaging, Hôpitaux Universitaires Paris Saclay, Assistance Publique-Hôpitaux de Paris, CHU Bicêtre, 94270 Le Kremlin-Bicêtre, France; duclos.valentin@icloud.com (V.D.); alexiep@ymail.com (A.I.); lea.gomez@aphp.fr (L.G.); 2Service Hospitalier Frédéric Joliot-CEA, 91401 Orsay, France; lucas.goldfarb@cea.fr; 3Université Paris Saclay, CEA, CNRS, Inserm, BioMaps, 91401 Orsay, France; 4School of Medicine, Université Paris Saclay, 94720 Le Kremlin-Bicêtre, France

**Keywords:** PET, quantification, molecular imaging, theranostics, multimodality, computer science

## Abstract

PET/CT molecular imaging has been imposed in clinical oncological practice over the past 20 years, driven by its two well-grounded foundations: quantification and radiolabeled molecular probe vectorization. From basic visual interpretation to more sophisticated full kinetic modeling, PET technology provides a unique opportunity to characterize various biological processes with different levels of analysis. In clinical practice, many efforts have been made during the last two decades to standardize image analyses at the international level, but advanced metrics are still under use in practice. In parallel, the integration of PET imaging with radionuclide therapy, also known as radiolabeled theranostics, has paved the way towards highly sensitive radionuclide-based precision medicine, with major breakthroughs emerging in neuroendocrine tumors and prostate cancer. PET imaging of tumor immunity and beyond is also emerging, emphasizing the unique capabilities of PET molecular imaging to constantly adapt to emerging oncological challenges. However, these new horizons face the growing complexity of multidimensional data. In the era of precision medicine, statistical and computer sciences are currently revolutionizing image-based decision making, paving the way for more holistic cancer molecular imaging analyses at the whole-body level.

## 1. Introduction

Awarded the medical invention of the year by Time Magazine in 2000, PET/CT molecular imaging has been imposed in clinical oncological practice over the past 20 years, substantially modifying the management of many cancer subtypes in daily practice [1,2,3]. Supported by the International Atomic Energy Agency, the Lancet Oncology commission on medical imaging and nuclear medicine—an international consortium established in 2018 to inventory and promote access to imaging and nuclear medicine for cancer care—has emphasized the substantial health and cost benefits of scaling up access to imaging and nuclear medicine for cancer care worldwide [4]. In particular, PET/CT and theranostics are now recommended imaging modalities for cancer care in tertiary health care centers.

Based on the well-grounded foundations of quantification and radiolabeled molecular probe vectorization, the goal of this article is to provide a holistic overview of the capabilities, current practice and emerging perspective of PET-molecular imaging for diagnosis, therapeutic evaluation and prognosis in clinical oncology.

## 2. PET Molecular Imaging: A Powerful Quantitative Imaging Tool with Different Levels of Analysis

PET imaging was mainly developed in the 1970s for brain research purposes. In the early 2000s, PET/CT hybrid imaging was rapidly imposed worldwide as a critical oncological imaging tool, paving the way for the molecular imaging-based assessment of tumors in clinical practice. During the last 20 years, constant technical improvements have led to a 3- to 5-fold decrease in injected radioactivity in clinical practice, making the debate on radiation exposure clearly outdated in view of the major benefit obtained in daily patient care. The recent FDA-cleared and CE-marked deep learning solutions that provide dose-reduced AI-based FDG PET/CT should optimize patients’ radiation exposure even more in the future (SubtlePET^TM^ AI [5]). With its very high detection sensitivity properties (picomolar, 10^6^ higher than standard morphological imaging), PET is a powerful imaging tool that can be used to quantify various biological processes. By quantification, one must understand a simple linear relationship between the numerical pixel value N measured in a region of interest (ROI) of the image and the biological radiotracer concentration [C] in the related tissue structure: N=k[C]. Importantly, this fundamental property depends on the control of many technical and physical factors to make an optimized and accurate link between the patient and its related optimized PET imaging data [6]. Regular quality controls, standardized practical procedures and CT-based attenuation corrections are among the most important prerequisites. Additionally, one must be aware of the impact of image reconstruction parameters on the measured PET data. For these reasons, the European Association of Nuclear Medicine (EANM) launched the EANM Research GmbH (EARL) initiative in 2006 to promote the standardization of PET practices (clinical and research) at the international level [7]. In the same way, and because molecular imaging is impacted by the patient’s physiological condition, several drug intakes, fasting procedures and delay times between the radiotracer injection and the image acquisition must be verified. Therefore, numerous international guidelines have been published to standardize these procedures [8,9,10]. For example, considering FDG PET imaging, the most widely used PET radiotracer in clinical oncology, and because tumor glucose metabolism increases with time [11], a fixed 60-min delay time between radiotracer injection and PET acquisition has been defined [8]. Additionally, metabolism-modifying drugs such as G-CSF and physical activity must be avoided, and a 6-h fast before and euglycemia at the time of image acquisition must be verified [8]. Based on these important technical considerations, several levels of quantification have been historically developed for PET imaging: from visual interpretation to full kinetic modeling, the level of information provided to the clinician is not the same and does not require the same acquisition procedures (Figure 1).

### 2.1. Visual Analysis

This first level of analysis only requires static PET imaging data (the standard of care in clinical practice): the radiotracer is injected intravenously into the patient, and a few minutes of PET acquisitions (typically 2 to 5 min per bed position, which is defined by the PET axial field of view) are started after a fixed delay time from the injection (e.g., 60 min for ^18^F-FDG). The resulting PET imaging data are, thus, an average snapshot of the entire acquisition time for each PET bed position (Figure 1). Visual analysis considers any non-physiological radiotracer uptake as abnormal. Although this protocol is very simple to use, accurate knowledge of radiotracer biodistributions is required (Figure 2). Additionally, this pattern-based approach is subjective and remains purely qualitative. To reduce the inter- and intraindividual variabilities, visual grade normalizations to reference regions have been proposed (cf infra). The diagnostic and prognostic capabilities of PET visual analyses over conventional imaging are perfectly illustrated in FDG-avid lymphoma patients [12], with reported management changes up to 45% at initial staging [13] and improved outcome prediction (both progression and event-free survival) at both early and end-of-treatment evaluations (for extensive review, please refer to Appendices A1–A3 of [12]). In the same way, other well-grounded illustrative foundations emphasizing the high suitability of visual-based PET assessments for the diagnosis or management of oncological patients include malignant melanoma [14], non-lepidic non-small-cell lung cancer (^18^F-FDG) [1,15,16], paraganglioma syndromes (^18^F-FDOPA for SDHC, SDHD and SDHAF2 gene mutations) [17,18,19] or, more recently, multiple myeloma [20].

### 2.2. Semi-Quantitative Analyses

To measure the biological aggressiveness of tumors and to monitor the response to treatment from standard-of-care static PET imaging data with more objective metrics, semi-quantitative approaches have been increasingly used in clinical studies. The standardized uptake value (SUV), the most widely used semi-quantitative PET metric that was first used in the 1980s [21], is a unitless parameter defined by the ratio of the time-decay-corrected radioactivity concentration measured in a region of interest ($C_{target},\;\mathrm{in}\;\mathrm{MBq}/\mathrm{mL})$ to the radioactivity injected in the body ($C_{injected}$, in MBq), normalized by the body weight (in g) or equivalent: $SUVbw=\frac{C_{target}}{C_{injected}/Weight}\times 1\;g/\mathrm{mL}$. SUV normalization to body surface area (SUV_bsa_) or, more recently, lean body mass (SUV_lbm_ or SUL) instead of body weight, has been proposed, and SUL is the current preferred metric for response assessment studies because of its lower dependency on body weight changes [8]. From this general concept, many SUV-based metrics have been reported since the 1990s, including SUV_max_ (the maximum pixel activity measured in the region of interest), SUV_mean_ (the mean of pixel activities measured in the region of interest) or SUV_peak_ (the mean of pixel activities measured in a volume of 1 cm^3^ centered on the higher uptake part of the region of interest), but also target background SUV ratios (SUV_r_) and composite whole-body scores, such as metabolic tumor volume (TV = $\sum_{1}^{n}\left(Volume_{n}\right)$ for *n* lesions identified at the whole-body level) or total lesion glycolysis (TLG = $\sum_{1}^{n}\left(SUV_{mean}n\times Volume_{n}\right)$ for *n* lesions identified at the whole-body level) (Figure 3). Importantly, all these SUV-based metrics intrinsically depend on numerous physical and technical factors [22,23], strongly limiting the absolute comparability at the multicenter level except if using proper standardization procedures [7]. Additionally, the semiautomated segmentation procedures used to generate metabolic volumes of interest are still mainly based on simple fixed thresholds in clinical practice (typically 3D isocontour at 41% or 50% of the maximum pixel value) [8], despite their well-known inaccuracy to delineate heterogeneous tumor lesions. The segmentation task is an entire research topic in itself, illustrating the richness of the many possibilities developed in this field [24]. In general, SUV-based metrics and derivatives have shown their relevance for early response assessments of treatment or prognosis in various malignancies, such as gastric cancer [25], rectal cancer [26], lymphoma [27,28,29,30], gastrointestinal stromal tumors (GISTs) [31,32] or multiple myeloma [33,34].

### 2.3. Kinetic Modeling

In fact, visual and semi-quantitative PET metrics derived from static PET images provide limited information. Using dynamic PET acquisitions provides more sophisticated metrics, allowing the quantification of advanced metabolic pathways at the cellular level (Figure 1). From Patlak simplified graphical analysis [35] to full compartmental analyses (1, 2, 3 or more compartments depending on the estimated biological function: perfusion, enzymatic activity, receptor binding) [36], PET kinetic modeling is considered the reference standard for absolute quantification. Compared to standard-of-care PET static measurements, dynamic PET metrics have provided promising results to improve the diagnosis, response assessment or prognosis of various malignancies [37]. To date, historical drawbacks have limited its use in clinical practice. In particular, the need for time-consuming acquisition schemes (up to one-hour continuous PET acquisition to estimate the full compartmental modeling of glucose metabolism [38]), the very limited axial coverage of the body extent in standard-of-care PET/CT devices and the complexity of image processing remain major limiting factors. However, the recent FDA clearances of large axial coverage and total body PET/CT systems [39,40,41], together with potential time-reduced acquisition procedures [42,43], improved reconstructions and postprocessing tools, will probably democratize dynamic whole-body PET in research and clinical practice, boosted by the growing molecular targeting requirements related to oncological drug development [44].

## 3. PET Molecular Imaging: Response Evaluation Criteria in Practice

The use of robust, easy-to-use and reliable interpretation criteria is of particular importance in oncological imaging. In clinical practice, many efforts have been made over the last two decades to standardize image analyses at an international level. Supported by historical RECIST morphological-based models, the response criteria are traditionally defined by four main categories: complete response (CR), partial response (PR), stable disease (SD) or progressive disease (PD). Although RECIST and its revised versions historically dominate this field in clinical trials [45], its limitations in predicting survival outcome or response to treatments and the massive deployment of PET imaging in worldwide oncological practice have motivated the emergence of PET-based response criteria in many cancer diseases. Depending of the targeted disease, several PET-based international criteria have been proposed as described in the next sections.

### 3.1. Solid Tumors 

A general historic of response criteria for solid tumor is provided in Table 1. In 1999, the European Organization for Research and Treatment of Cancer (EORTC) first introduced metabolic information from ^18^F FDG-PET imaging (namely, SUV_max_) into the response assessment criteria of oncological diseases [46]. In practice, the EORTC criteria remained in the background of RECIST until 2009 because of the well-known inherent inter-center variability of the SUV_max_ PET metric [47]. In 2009, a very large step was made with the introduction of the PET-Response Criteria in Solid Tumors (PERCIST) [48]. For the first time, a critical effort has been made to propose a well-defined, reliable and robust standardized PET methodology in this field (Figure 4). To overcome the limitations of SUV_max_, SUV_peak_ was introduced, together with SUV normalization by lean body mass (SUL) and a precise explanation of the definition and number of measurable targeted tumor lesions. Additionally, the percent change in SUL_peak_ between the two examinations was set to 30% to integrate the inter-center variabilities. The PERCIST criteria take full interest, particularly in the assessment of cytostatic chemotherapies, as they can demonstrate metabolic changes when no anatomical changes are observed. To date, PERCIST has surpassed RECIST in numerous cancer diseases for the prediction of patient outcome or the assessment of responses to treatment in many cancer diseases, emphasizing the relevance of this powerful PET-based evaluation tool. In particular, breast cancer [49], esophageal cancer [50], Ewing sarcoma [51] and non-small-cell lung cancer (NSCLC) [52,53] have been investigated for this purpose. With the emergence of immunotherapy, standard imaging criteria (both morphological and metabolic-based) have rapidly required refinements to integrate atypical response patterns such as pseudoprogression or hyperprogression diseases, which can lead to significant misclassification of patients [54]. To mirror the morphological definition of two new categories of response assessment to treatment in the dedicated iRECIST update (namely, unconfirmed and confirmed progression diseases), iPERCIST criteria were recently proposed and tested retrospectively in a study of 28 cases of NSCLC being treated with nivolumab [55]. Previously, PET/CT Criteria for Early Prediction of Response to Immune Checkpoint Inhibitor Therapy (PECRIT) have been assessed prospectively in advanced melanoma under immune checkpoint inhibitors [56]. Combining anatomical and functional imaging data to predict eventual response to ICI reached 100% sensitivity, 93% specificity and 95% accuracy. However, these findings are mainly limited by the small number of patients evaluated and require validation in larger cohorts. More recently, Anwar H. and coworkers proposed the PET Response Evaluation Criteria for Immunotherapy (PERCIMT) in 41 melanoma patients [57,58]. Interestingly, PERCIMT criteria consider the number of new lesions and their extent during therapy, which allows better patient stratification compared to standard SUV-based parameters. The new appearance of ≥4 metabolically active lesions with functional diameters <1.0 cm or ≥3 lesions >1.0 cm in diameter seemed correlated with real progression rather than pseudoprogression. At the same time, ^18^F-FDG PET imaging offers a unique opportunity to detect immune-related side effects at the whole-body level, such as thyroiditis, gastritis, colitis, pneumonitis, sarcoidosis [59] and polymyalgia rheumatica-like syndromes [60]. Interestingly, few studies have reported an association between immune activation [61] or immune-related side effects assessed by PET and patient response [62,63]. Although these recent results must be confirmed in larger prospective studies, metabolic-based imaging appears to be a promising tool in this field.

### 3.2. Lymphoma

In Hodgkin Lymphoma (HL) and Non-Hodgkin Lymphoma (NHL, mainly Diffuse large B-cell lymphoma DLBCL), given the natural high avidity of aggressive tumor cells for ^18^F-FDG and the binary metabolic response to treatment, dedicated PET response evaluation criteria have progressively emerged to currently become the most powerful imaging tool in this field (Table 2). Considering the extensive evidence-based literature of the first decade of the 21st century, PET was integrated into the international guidelines for response assessment of lymphoma in 2007 [64], a revised version of the historical international working group morphological criteria [65]. Seven years later, the Lugano criteria refined these guidelines [66], confirming the role of PET as a key imaging modality in the diagnosis and response assessment of lymphoma. PET imaging was definitively integrated into the Ann Arbor staging system (Lugano staging classification), and the PET Deauville 5-point visual grading scale was defined as the main response evaluation tool (in standard therapeutic schemes, end-treatment scores of 1, 2 or 3 were considered CR, whereas scores of 4 and 5 were considered non-responses, as illustrated in Figure 5). Additionally, based on the relevance of PET in this field, bone marrow biopsy is no longer indicated for the assessment of bone marrow involvement in HL and is only required in negative PET cases with consequences for patient management in DLBCL. To increase the reliability of the procedure and improve the prognostic value of PET imaging, interim SUV-based semiquantitative metrics are currently being extensively investigated [28,67,68]. In particular, a very recent large prospective study with 158 new DLBCL patients reported higher prognostic value for interim SUV-based metrics (interim ΔSUV_max_ = percent change between baseline PET and PET performed after two cycles of treatment) compared to the standard visual grading scale evaluation [69]. In particular, only the interim ΔSUV_max_ predicted both PFS and OS. These findings and those of other recent clinical trials highlight the relevance of using more quantitative PET methods in this field [70,71]. Finally, for solid tumors, PET imaging criteria were recently adapted to immunotherapy by integrating indeterminate response features: the lymphoma response to immunomodulatory therapy criteria (LYRIC) in 2016 [72] and the Response Evaluation Criteria in Lymphoma (RECIL) in 2017, with more morphological parameter weighting [73].

### 3.3. Multiple Myeloma

Magnetic resonance imaging (MRI) is considered the gold standard to detect bone marrow involvement in multiple myeloma due to its very high anatomical resolution and soft tissue contrast. However, its limited value in assessing the response to treatment and the time-consuming whole-body acquisition protocols have promoted the use of ^18^F-FDG PET/CT as a fast-whole-body complement imaging modality to manage patients with MGUS and multiple myeloma. In comparison with MRI, ^18^F-FDG PET/CT shows faster normalization of image findings [74]. Recently, the International Myeloma Working Group has integrated ^18^F-FDG PET/CT into the diagnostic criteria of multiple myeloma, considering abnormal bone ^18^F-FDG uptake as a C.R.A.B feature (C = hypercalcemia; R = renal failure; A = anemia; and B = bone lesions) [75]. Although clinically promising in this field, several drawbacks make the interpretation of ^18^F-FDG PET challenging in multiple myeloma, especially at the time of initial diagnosis: as a heterogeneous disease, multiple myeloma can present with variable metabolic uptake, ranging from low to extremely high; additionally, anemia and G-CSF may stimulate BM uptake, reducing the detectability of axial targets. All these challenging aspects and pitfalls require extensive experience and knowledge of the particular aspects of MM. In past years, different interpretation criteria have been proposed. Some groups have proposed semiquantitative SUV-based parameters to assess the disease burden [76]. Recently, a consortium of nuclear medicine experts, hematologists and medical physicists proposed standardized criteria to promote the use of PET in clinical trials: the Italian myeloma criteria for PET use (IMPeTUs). These new criteria, combining the Deauville five-point scale and morphological features (site and number of lytic lesions, Table 3) [77], have provided high reproducibility and can be considered a basis for harmonizing PET interpretation in multiple myeloma. Although recent findings showed the prognostic value of these criteria for both OS and DFS [20], further prospective clinical trials are warranted to confirm the relevance of IMPETUS in this field.

## 4. PET Molecular Imaging: The Promising Clinical Perspective of Radioligand Molecular Imaging and Therapy

PET imaging detection is conceptually based on the radionuclide labeling of molecular probes. Considering several intrinsic chemical limitations of vector-radioligand coupling, PET molecular imaging provides almost unlimited opportunities to map numerous physiological or pathophysiological targeted processes at the whole-body level, with picomolar detection sensitivity (Figure 6). In the last decade, the integration of PET imaging with radionuclide therapy, also known as radiolabeled theranostics, has paved the way towards highly sensitive radionuclide-based precision medicine. In particular, major breakthroughs are currently expected in two clinical fields: neuroendocrine tumors (NETs) and prostate cancer (PCa).

### 4.1. NETs

According to the Surveillance Research Program of the National Cancer Institute [78], NETs account for only 0.49% of all malignancies. However, an unexplained increased incidence has been observed over the past decades. The emergence of somatostatin receptor-based PET diagnostic imaging revolutionized patient care in the late 1980s, for whom strategy management was very limited [79,80]. A further step was reached with the emergence of PET-based somatostatin receptor-imaging in the 2000s, combining the inherent higher physical properties of PET systems over conventional scintigraphy (spatial resolution, sensitivity detection, unbiased quantification), improved kinetics (faster clearance and tissue penetration with the last generations of targeted radioligands) and chemical properties (chelation-based, allowing fast and rapid vectorized radioligand switching from diagnosis with ^68^Ga to therapy with ^90^Y, ^177^Lu or ^225^Ac) [81,82,83]. Based on 2105 patients, the pooled diagnostic performance of SSR-PET from 22 studies provided a sensitivity and specificity of 93% (95% CI 91–94%) and 96% (95% CI 95–98%), respectively [84], surpassing conventional scintigraphy in this field [85]. Such detection performance led to the integration of SSR-PET into the international strategy of patient management, from initial staging to recurrence and palliative care for almost all NETs [86,87] (Figure 7). Recently, the phase III NeuroEndocrine Tumors Therapy clinical trial (NETTER-1) validated the use of SSR-based radioligand therapy in well-differentiated metastatic NETs of the midgut [88]: ^177^Lu-SSR showed a 79% reduction in risk of progression and an estimated PFS of 40 months compared to 8.4 months for high-dose octreotide therapy (79% lower risk of disease progression or death in the 177Lu-Dotatate group). For undifferentiated high-grade NETs, SSR-based radioligand therapy is still debated [89,90]. Some findings suggest that aggressive and higher grade 2-3 NETs but still well-differentiated with sufficient expression of somatostatin receptor to be visualized through prognostic SSR-PET could benefit from SSR-based radioligand therapy [91]. The ongoing NETTER-2 clinical trial (registered as NCT03972488) should answer this question. Finally, beyond SSR-PET, standard FDG PET may be used as a complementary tumor burden mapping in the patient management strategy due to its powerful independent prognostic value [92,93].

### 4.2. PCa

Prostate cancer is the second most frequent malignancy in men worldwide [94]. Because of the limited value of standard imaging, choline PET (a membrane phospholipid radiolabeled either with ^11^C or ^18^F) was widely used in the 2000s for the early assessment of biochemically recurrent prostate cancer. Extensive literature has shown choline PET to be of high relevance for the detection of recurrent lymph nodes or bone metastases in treated patients, especially with PSA above 2 ng/mL or high PSA velocity/doubling time [95,96,97]. However, poorer detection rates for PSA under 2 ng/mL and low specificity for treatment-naïve patients motivated the development of more specific molecular probes in this field. In this context, prostate-specific membrane antigen (PSMA), a transmembrane protein of unclear biological function that is overexpressed 100- to 10,000-fold in PCa cells compared to normal tissues [98,99,100], has gained increasing interest in the last decade. Given its high sensitivity for the detection of PCa lesions, PSMA radiolabeling is currently changing the methods for managing early biochemically recurrent PCa and occult metastatic PCa, for which conventional imaging modalities (MRI and CT) lack sensitivity and specificity [101]. Except for neuroendocrine PCa [102], PSMA expression increases with tumor dedifferentiation and in metastatic castration-resistant prostate cancer. A recent pooled analysis of 5113 patients (43 studies) imaged with ^68^Ga PSMA PET showed overall detection rates of 94% at PSA levels above 2.0 ng/mL for biochemically recurrent PCa treated by radical prostatectomy, with higher detection rates than conventional imaging at PSA levels under 0.5 ng/mL [103]. Moreover, numerous studies have shown that PSMA PET radiotracers surpasses the actual concurrence (choline or fluciclovine PET radiotracers) to detect biochemically recurrent PCa, especially at PSA levels under 1 ng/mL (for reviews, please refer to [104,105]). Beyond recurrent disease management and given the growing literature in this field, especially the very good agreement with histology [106,107], PSMA PET has been rapidly imposed for the primary management of PCa patients [108,109,110]. In December 2020, the US Food and Drug administration granted the first clinical approval for institutional use (University of California, UCLA and USCF) of ^68^Ga PSMA PET (PSMA-11) for the initial staging and the detection of recurrence in PCa patients based on recent clinical trial findings [111,112,113,114]. This milestone achievement should pave the way to a new standard of care for patients with PCa. From this perspective, and for simpler availability and production workflows, several diagnostic ^18^F-radiolabeled PSMA PET tracers have already been integrated into the research pipeline [115,116,117,118]. Together with these recent imaging advances, PSMA radioligand therapy has also emerged in patients with metastatic castration-resistant PCa (Figure 8). Mainly based on small-molecule inhibitors of PSMA (PSMA-I&T, PSMA-617), 10 large phase I-III PSMA RLT trials are currently ongoing and are using either ^177^Lu or ^225^Ac radionuclides (for details please refer to [119]). Among the promising literature findings already published, two prospective clinical trials are considered a very large step in this field: the LuPSMA trial, a single-arm phase 2 study that assessed the safety and efficacy of ^177^Lu-PSMA-617 in 30 patients with metastatic castration-resistant PCa [120], and the TheraP multicenter randomized open-label phase 2 trial, which showed a decrease in PSA levels of at least 50% from baseline for the ^177^Lu-PSMA-617 group compared to the cabazitaxel group, along with longer PFS (HR 0.63, *p* = 0.0028), lower toxicity events and higher pain improvement [121]. Finally, the very recent first historical standardized reporting guideline for PET imaging in this field emphasizes the growing importance of PSMA-based radiotracers in the management of PCa and the strong motivation of the international nuclear imaging community to accelerate its use in future clinical trials [122].

### 4.3. Other Near-Future Promising Perspectives in Clinical Practice: Immuno-PET and Beyond

Beyond the emerging success of NETs and PCa radioligand targeting, various molecular cancer targets have been developed since the historical use of ^18^F-FDG in the late 1980s [123], based on the fundamental aerobic glycolytic properties of cancer cells, also known as the “Warburg effect” [124]. Emphasizing the unique capability of PET molecular imaging to constantly adapt to emerging oncological challenges, other biological properties of cancer have been assessed in numerous phase 0–3 trials, including hypoxia, apoptosis or protein synthesis (for review please refer to [125]). The hallmark capabilities of cancer are an evolving concept [126], and tumor immunity (tumor cells and the interlinks with their immune microenvironment) is currently a very hot topic of interest, given the revolution provided by immunotherapy checkpoint inhibitors [127,128]. Despite progressing results since the first FDA approval in 2011, the response rate to checkpoint inhibitors remains approximately 13% of eligible patients [129]. Many efforts are currently made to better identify patients who would be eligible for these new drugs [130,131]. Additionally, the numerous updates of limited current imaging criteria illustrate the needs of new tools to better characterize the immune ecosystem of tumors to improve treatment strategies [132,133]. Immune checkpoint-targeted radiolabeled monoclonal antibodies (IC-PET) and fibroblast activation protein-targeted radiolabeled inhibitors (FAPi-PET) are two serious candidates in this field. In the last five years, several dozen preclinical IC-PET studies with ^64^Cu, ^89^Zr, ^68^Ga, ^124^I or ^18^F radionuclides have been published, but numerous challenging technical considerations remain to reach optimal IC-PET pharmacokinetics and biodistribution in practice (size, Fc-mediated functions and charge, radiolabeling strategy) (Figure 9); for a review, please refer to [134]. Two very recent first-in-human studies assessed its clinical safety and feasibility [135,136]. The results from the study by Niemeijer and coworkers showed that for 13 patients with advanced NSCLC (follow-up of 3 months), IC-PET (^89^Zr-nivolumab) findings correlated with immunohistochemistry [136]. In the study by Bensch F and coworkers, who included 25 patients with various cancer subtypes (bladder, NSCLC, triple negative breast cancers; median follow-up of 21.9 months), IC-PET (^89^Zr-atezolizumab) better correlated with PFS and OS than immunohistochemistry or RNA-sequencing [135]. These very promising results have promoted the launch of numerous IC-PET clinical trials focusing on NSCLC, HNSCC, lymphomas, RCC, breast cancers and melanomas (for a review please refer to [134]). Finally, FAPi-PET has recently emerged as a PET molecule targeting cancer-associated fibroblasts (CAFs) [137,138,139]. CAFs play critical roles in tumor progression and immunity regulation and represent a promising therapeutic target [140]. Activated CAFs highly express FAP, a glycoprotein enzyme with peptidase activity [141]. Combining both PET molecular probes of tumor cells (^18^F-FDG PET) and their surrounding stroma (FAPi PET) provides an open exciting perspective, as illustrated by recent findings in oncological patients with inconclusive ^18^F-FDG PET findings [142], or to optimize the tumor volume delineation for radiotherapy planning [143]. Together with imaging capabilities, FAP-radioligand therapy is also under development [144,145], paving the way towards stroma-targeted radiolabeled theranostics.

## 5. PET Molecular Imaging: The Perspectives of Statistical and Computer Sciences for Multidimensional Image-Based Decision Making

In the last 20 years, the concept of tumor heterogeneity—where a tumor mass is considered a patchwork of distinct genotypic and phenotypic subcellular populations—has been imposed as a strong factor of treatment resistance [146,147]. In this context, the unsuitability of standard image-based criteria to accurately assess the response to nonconventional therapies [148,149], together with the rapid emergence of computer science in the field of medical imaging, promoted the development of radiomics—the extraction of high-throughput quantitative metrics from medical images—which is a large-scale image-based approach derived from OMICS (genomics, transcriptomics, proteomics, metabolomics, etc.) of which the purpose is to better capture tumor heterogeneity from standard-of-care medical images to build relevant diagnostic or predictive models [150]. Despite promising FDG PET results in the early 2000s [151,152] and efforts in recent years to propose “user-friendly” dedicated software for clinical use, radiomics has not yet reached a sufficient level of relevance in clinical practice [153]. In essence, the standard radiomics pipeline integrates image acquisition, segmentation and standardization, handcrafted feature extraction (intensity, shape and texture parameters) and selection through machine learning-based statistical analyses to build optimized diagnosis and prognosis endpoint models. One major drawback of such a huge operational processing pipeline remains its multilevel subjectivity and technical dependencies [154]. A recent systematic review based on 624 records (41 full-text articles mixing lung, head and neck, esophageal, rectal, cervical and breast cancers, mainly describing CT and PET findings) specifically addressed the repeatability and reliability of radiomic features and reported probable high variability at each main processing step for the majority of second- (shape-based metrics) and third-order (texture-based metrics) feature classes [155]. Nonetheless, this field of research is maturing based on years of hindsight. In 2020, the Imaging Biomarkers Standardization Initiative (IBSI), a task force of 20 research groups, established international standardized frameworks for radiomics computation and validated the use of 164 PET radiomics features [156]. Another major issue concerns the use of unsuitable statistical procedures in the vast majority of reported radiomics studies, probably partly explained by two factors: the strong difficulty of collecting and structuring large homogeneous datasets in “real-life” clinical practice [157] and the lack of knowledge in machine learning/data mining, a “new” field of skills to be learned by nuclear medicine practitioners. Again, the practice is maturing. Harmonization procedures have emerged to facilitate multicenter studies in practice and remove the so-called “center effect” [158,159]. Additionally, several statistical rules of thumb have gradually been imposed on the community, improving the development and validation of predictive models, particularly the use of a sufficient number of patients per radiomics feature, the dataset partitioning into independent training, validation and test subsets, the hyperparameter optimization and cross-validation of the algorithms and the use of objective performance metrics [160,161,162]. In the era of precision medicine, multiparametric imaging offers unique opportunities to characterize tumor behavior at an advanced multidimensional imaging level [163,164]. Interestingly, multiparametric radiomics has recently been shown to surpass single-modality procedures. In their proof-of-concept study, Vallières M. and coworkers demonstrated the superiority of joint FDG PET and MRI texture features to predict the risk of lung metastases in soft tissue sarcomas [165]: in a cohort of 51 biopsy-proven sarcomas, first- and second-order radiomics features were extracted from either FDG PET, MRI (T1- and T2-weighted) or fused PET/MRI baseline data. Among all the multivariable predictive models tested, the models constructed from fused PET/MRI data surpassed those generated either from PET or PET + MRI separated scans (AUC of 0.98 ± 0.002, sensitivity of 0.955 ± 0.006, specificity of 0.926 ± 0.004). More recently, Mu W. and coworkers compared the predictive value of single-modality and fused FDG PET/CT-based radiomic signatures (baseline scans) to predict the durable clinical benefit of immune checkpoint inhibitors in advanced NSCLC patients [166]. Trained on 99 patients and validated on two independent cohorts of 47 NSCLC (retrospective) and 48 NSCLC (prospective) patients, the predictive models, which included features generated from fused PET/CT data, were improved in all the training and test sets, reaching AUCs of 0.86 (95% CI 0.79–0.94), 0.83 (95% CI 0.71–0.94) and 0.81 (95% CI 0.68–0.92), respectively.

By capturing the pooled radiomic signatures of tumors across different imaging modalities, multiparametric radiomics changes the dimensionality of analyses: the extracted features no longer represent intervoxel relationships in a modality of interest but the multidimensional tumor behavior, driven by the properties of each modality. Such a paradigm shift will probably be enhanced with fully integrated PET/MRI systems, which have been clinically available since the beginning of the 2010s [167]. In this way, rather than using standard-of-care qualitative or semiquantitative PET, CT or MRI images, one could move towards more quantitative multiparametric analyses in future practice [168]. Moreover, fully automated and miniaturized production systems for PET radiotracer use in a “dose-on-demand” mode could stimulate multi-PET probe patient imaging in future clinical practice (iMiGiNE project, PMB-Alcen/SIGMAPHI/CEA).

To face the growing complexity of this arsenal of multidimensional data, the use of deep learning—a branch of machine learning using neural networks (convolutional neural networks (CNNs) in image processing)—appears to be a promising way to assist practitioners in the future. Because CNNs learn characteristics directly from raw images, feature extraction and prediction tasks can be performed jointly in an embedded process, bypassing classical handcrafted segmentation, feature selection or classification multilevel steps [169,170]. Moreover, two recent studies nicely illustrate the potential gain at the whole-body level of automated CNN-based detection and classification tasks in PET/CT. From 629 FDG PET/CT data (302 lung cancer and 327 lymphoma) retrospectively labeled by two nuclear medicine physicians, Sibille L. et al. trained, validated and tested a CNN research prototype (PET Assisted Reporting System PARS, Siemens Healthineers) to automatically detect and classify abnormalities (anatomic location, suspicious or nonsuspicious), reaching a classification AUC of 0.988 (95% CI 0.982–0.994) [171]. In their study, including almost 3500 FDG PET/CT data points from patients with various malignancies/benign diseases, Kawauchi K. et al. successfully classified PET data into three predefined classes (benign, malignant, equivocal), and their CNN reached accuracies of 99.4, 99.4 and 87.5% at the patient level [172]. Beyond evident improvements in the entire image processing workflow, numerous technical challenges will have to be faced before clinical validation. As an example, the recent external validation of the PET Assisted Reporting System (PARS, Siemens Healthineers) [171] on two French cohorts (cohort 1: 119 cases of DLBCL; cohort 2: 430 miscellaneous cancers) showed discrepancies between the reference manual and CNN-based procedures (cohort 1: median Dice score = 0.65; ICC_TMTV_ = 0.68; cohort 2: median Dice score = 0.48; ICC_TMTV_ = 0.61) [173]. For multidimensional image-based decision making, the level of image dimensionality (how many modalities? how many parameters? what level of quantification?) and the optimal way to integrate this multidimensionality in the depths of CNNs [174] will also require many research investigations. Regardless, deep learning-based multidimensional quantification can pave the way for more operational and holistic cancer molecular imaging analyses at the whole-body level (Figure 10).

## 6. Conclusions

PET molecular imaging is a powerful imaging modality for quantifying tumor processes. After 20 years of clinical use around the world, PET/CT has proven its unique value for the diagnosis and therapeutic evaluation of many cancers. This maturity now allows us to consider a move towards more sophisticated levels of analyses in future clinical practice, boosted by the development of PET-based multimodal imaging, multiprobe radiolabeling and multidimensional data mining. A more holistic image-based representation of oncological processes becomes conceivable at the whole-body level, making PET molecular imaging a tool of choice in precision medicine for the next few years.

## Figures and Tables

**Figure 1 ijms-22-04159-f001:**
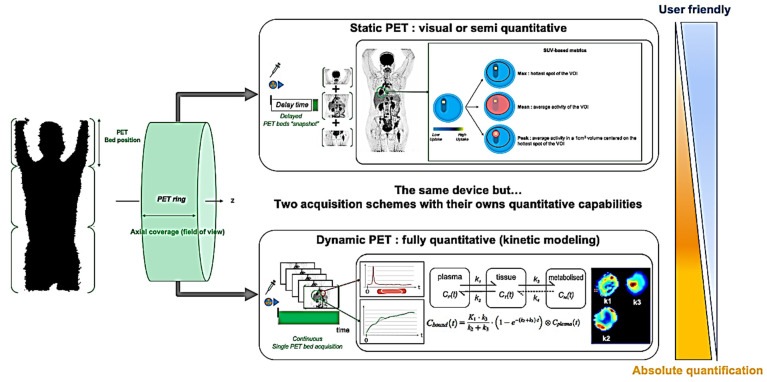
Main concepts of image acquisition and quantification in PET. The image coverage (PET bed position) depends on the axial field of view of the PET ring. Two acquisition schemes may be performed: the static mode (top panel), massively used in current practice, consists of acquiring snapshots of each PET bed position successively (a few minutes per bed position) at a fixed delay time after radiotracer injection (60 min in oncology), and the dynamic mode (bottom panel), currently devoted mainly to research purposes, consists of acquiring a PET bed position continuously (from a few minutes to one hour depending on the mathematical model used). The arterial input function (red curve) and lesion time-activity curve (green curve) are measured from volumes of interest, fitted to mathematical models to generate parametric maps of the lesion of biological significance (perfusion, enzymatic activity, metabolic rate of glucose consumption, etc.). While the static acquisition scheme is fast and user friendly, only visual and SUV-based semiquantitative metrics can be generated. However, dynamic PET models provide advanced quantitative metrics of sophisticated biological significance and remain the absolute reference standard for quantification.

**Figure 2 ijms-22-04159-f002:**
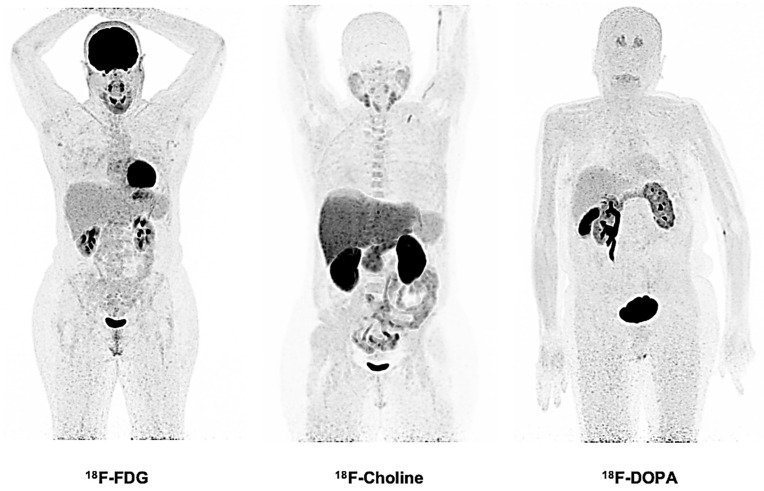
Normal biodistributions of three widely used PET radiotracers in oncological practice (static PET acquisitions): ^18^F-FDG to assess glucose metabolism, ^18^F-choline to assess membrane renewal and ^18^F-FDOPA to assess the metabolism of L-DOPA. In the three cases, the radiotracer physiological biodistributions have their own characteristics.

**Figure 3 ijms-22-04159-f003:**
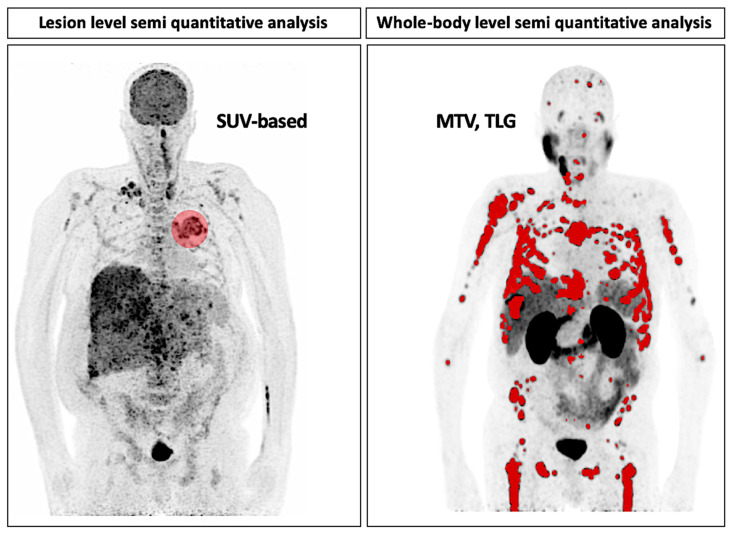
Semiquantitative PET metrics can be extracted at the lesion level (left panel, ^18^F-FDG PET scan of a patient with NSCLC) or at the whole-body level (right panel, ^68^Ga-PSMA PET scan of a patient with prostate cancer, courtesy of Jérémie Calais and Andrei Gafita, Ahmanson Translational Theranostics Division, UCLA, Los Angeles, CA, USA). For whole body metrics, the sum of each SUV-based metric extracted at the lesion level is used: metabolic tumor volume (MTV) or total volume (TV) is defined at a fixed SUV threshold (typically 40% or 50% of SUV_max_ in standard practice), and total lesion glycolysis (TLG) is defined as MTV
× SUV_mean_. MTV and TLG are surrogates of tumor burden at the whole-body level.

**Figure 4 ijms-22-04159-f004:**
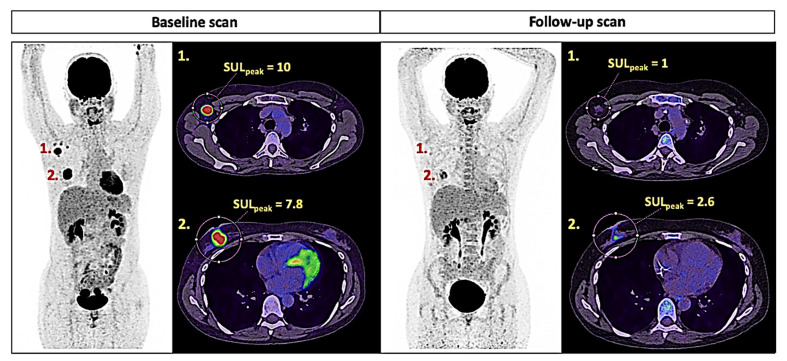
PERCIST tumor response assessment (breast cancer with N+ extension, ^18^F-FDG PET). The SUL_peak_ of the targets at baseline and follow-up, measured semiautomatically from the VOI on static PET images, provided between-scan differences of ΔSUL_peak_ = −90% and −67%, respectively, for lesions 1 and 2. According to PERCIST criteria, it is considered a partial metabolic response (ΔSUL_peak_ > −30%).

**Figure 5 ijms-22-04159-f005:**
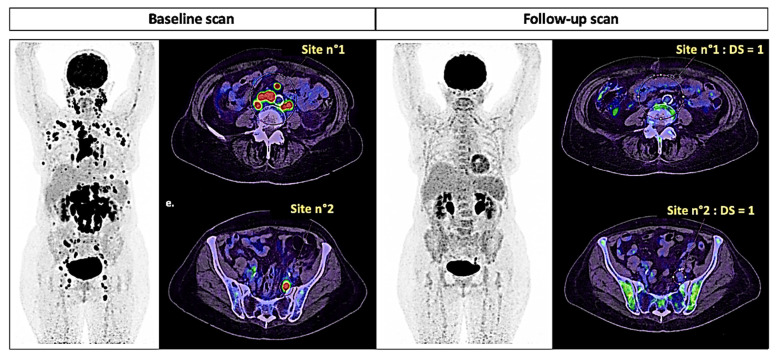
Lugano response assessment for lymphoma (DLBCL stage IV Ann Arbor, ^18^F-FDG PET). Visual grading assessment at follow-up corresponds to a Deauville score of 1 (DS = 1, no visual uptake). According to the Lugano criteria, it is considered a complete metabolic response.

**Figure 6 ijms-22-04159-f006:**
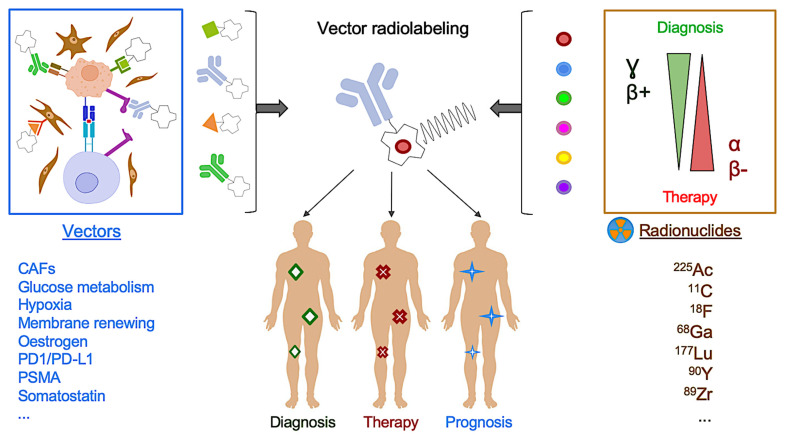
Principles of vectorized PET molecular imaging. Numerous relevant molecular probes constitute a powerful arsenal to characterize tumor biological processes in vivo. Vector radiolabeling by diagnostic radionuclides, in particular PET radionuclides (β^+^ emitters), allows tumor molecular targeted mapping (quantification at the whole-body level for diagnosis/prognosis and monitoring purposes). For the same vector, switching from diagnostic to therapeutic radiolabeling (either α or β^−^), so-called theranostics, allows vectorized internal radiation therapy with precise characterization of the tumor burden biodistribution. Therapeutic radionuclides have their own properties. With their very short radiation range of 47–85 μm, high energy α-emitters (^225^Ac) provide limited off-targeted irradiations with a high local cytotoxic effect, regardless of the cycle phase or oxygenation status. Importantly, these emitters require a high degree of target internalization. Currently, their use is limited by the worldwide production capabilities. Of lower energy and better available in practice, β^−^ emitters (^177^Lu, ^90^Y) provide better penetration ranges (several mm), which is of particular interest in the case of high targeted or heterogenous volumes, but induces higher hematological and renal toxicities.

**Figure 7 ijms-22-04159-f007:**
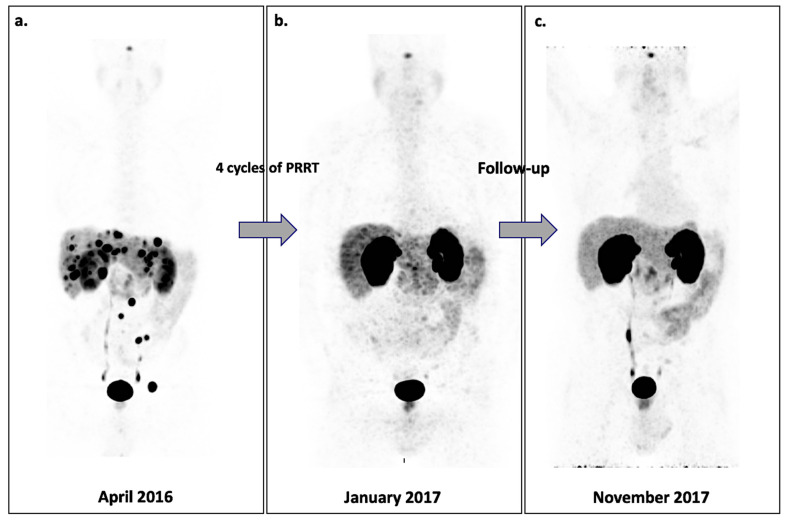
Theranostics and Peptide Receptor Radionuclide Therapy (PRRT). A patient with a NET diagnosed in 2015 (T3N0M1, G2, 20% Ki67) who was treated by surgery (distal pancreatectomy, splenectomy and liver resection) and radiofrequency ablation (liver metastasis). Somatostatin receptor (SSTR) PET imaging performed with ^68^Ga-DOTATOC before PRRT in April 2016 was positive, showing disseminated liver and bone lesions with intense uptake (Krenning score = 4) (**a**). After ^177^Lu-octreotide, all the foci disappeared (Krenning score = 0) on posttreatment (**b**) and follow-up ^68^Ga-DOTATOC PET (**c**), illustrating a complete response. Images courtesy of Jérémie Calais and Martin Auerbach (Ahmanson Translational Theranostics Division, UCLA, Los Angeles, CA, USA).

**Figure 8 ijms-22-04159-f008:**
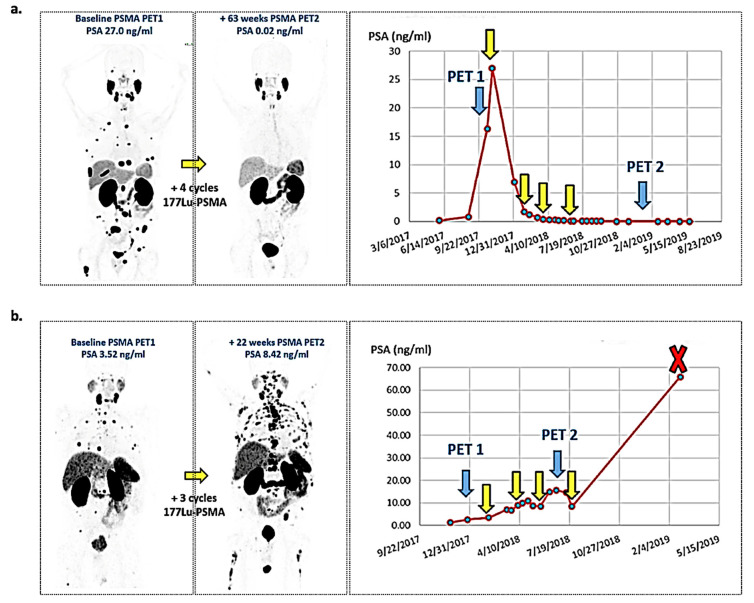
Theranostics and PSMA radionuclide therapy. (**a**) Example of a patient with PCa with complete response after treatment with ^177^Lu-PSMA-617: all the lesions disappeared on the follow-up ^68^Ga-PSMA PET at one year, together with the PSA level. (**b**) Despite the impressive successes reported with ^177^Lu-PSMA, several problems remain unresolved, as illustrated in this nonresponder case, with significant progression on the follow-up ^68^Ga-PSMA PET, together with the rising curve of PSA after treatment. Images courtesy of Jérémie Calais (Ahmanson Translational Theranostics Division, UCLA, Los Angeles, CA, USA).

**Figure 9 ijms-22-04159-f009:**
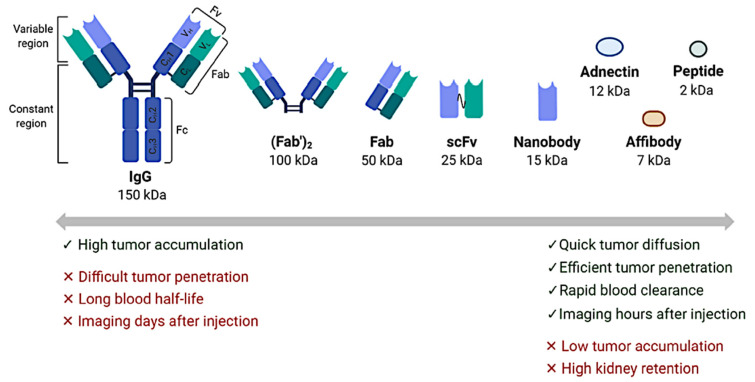
Different radioligand formats for in vivo PET imaging. A schematic representation of various radioligand formats with distinct pharmacokinetic properties that can be used for PET imaging of immune checkpoints: intact IgG antibody; antibody-derived fragments, including (Fab’)2, Fab, scFv and nanobody; and smaller protein scaffolds (adnectin, affibody, small peptide). Abbreviations: VL: light-chain variable domain; VH: heavy-chain variable domain; CL: light-chain constant domain; CH: heavy-chain constant domain. Image and legend courtesy of Alizée Bouleau/Charles Truillet/Vincent Lebon (Université Paris Saclay, CEA, CNRS, Inserm, BioMaps) [134].

**Figure 10 ijms-22-04159-f010:**
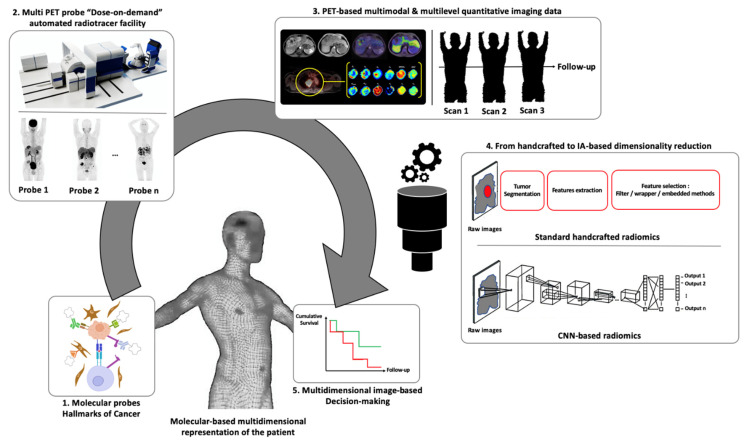
General overview of the potential of molecular-based multimodal imaging in the era of precision medicine: (1) Molecular probes are defined based on cancer hallmark knowledge. (2) Miniaturized automated PET radioligand production facilities could allow very fast and easy PET probe radiolabeling in future clinical practice, stimulating multi-probe PET imaging of oncological patients. (3) Static and/or dynamic multimodal PET-based acquisition procedures generate multidimensional image-based metrics at baseline and during follow-up. (4) Applications of statistical and computer sciences would facilitate the extraction and dimensionality reduction of all this information to make optimized image-based decision models (5) The panel 2 subpart of the prototypal “Dose-on-Demand” automated radiotracer facility is courtesy of Ludovic Le Meunier (PMB-Alcen, iMiGiNE project, PMB-Alcen/SIGMAPHI/CEA).

**Table 1 ijms-22-04159-t001:** Response evaluation criteria in solid tumors.

	Response to Treatment			
Criteria	CR	PR	SD	PD
** EORTC (1999) **	Reduction of ^18^F-FDG uptake to background	≥15% reduction of ^18^F-FDG uptake	Neither CR, PR nor PD	≥25% increase in ^18^F-FDG uptake
** PERCIST (2009) **	Reduction of ^18^F-FDG uptake to background	≥ 30% reduction in SULpeak	Neither CR, PR nor PD	>30% increase in SUL peak
** PECRIT (2017) **	Metabolic disappearance of target; SAD reduction of LN	≥30% reduction in SULpeak ≥30% decrease in TL diameter sum	Neither CR, PR nor PD	>30% increase in SUL peak Or new metabolically active lesion ≥20% increase in TL diameter Or new lesions
** PERCIMT (2018) **	Complete resolution of all ^18^F-FDG-avid lesions	Complete resolution of some ^18^F-FDG- lesions	Neither CR, PR nor PD	≥4 new lesions ≤ 10 mm Or ≥3 new lesions > 10 mm Or ≥2 new lesions > 15 mm
** iPERCIST (2019) **	Complete resolution of ^18^F-FDG uptake	≥30% decrease in the target tumor ^18^F-FDG	Neither CR, PR nor PD	≥30% ^18^F-FDG uptake Or new ^18^F-FDG target (UPMD). Need second PET at 4–8 weeks later (CPMD); if progression is followed by PMR or SMD, the bar is reset.

CR: Complete response, CT: Computed Tomography, ^18^F-FDG: ^18^F-Fluorodeoxyglucose, PD: Progressive Disease, PET: Positron emission tomography, PR: Partial response, SAD: Short-Axis Diameter, SUL: SUV normalized by lean body mass, TL: Target Lesions, UPMD: Unconfirmed Progressive Metabolic Disease, CPMD: Confirmed Progressive Metabolic Disease, PMR: Partial Metabolic Response, SMD: Stable Metabolic Disease.

**Table 2 ijms-22-04159-t002:** Response evaluation criteria in lymphoma.

	Response to Treatment			
Criteria	CR	PR	SD	PD
** LUGANO (2014) **	CT: reduction of lesions to normal size PET: normalized ^18^F-FDG-uptake (DS 1-3)	CT: ≥ 50% reduction in SPD of up to 6 lesions PET: reduced 18F-FDG-uptake (DS 4-5)	CT: neither sufficient change for PD nor PR PET: unchanged 18F-FDG-uptake (DS 4-5)	CT: ≥ 50% increase in SPD of lesions New lesion(s) PET: increased 18F-FDG-uptake (DS 4-5) or new 18F-FDG-avid lesions
** LYRIC (2016) **	Same as Lugano	Same as Lugano	Same as Lugano	Adapted from Lugano to indeterminate response (IR) categories: IR1: ≥ 50% increase in SPD in 12 weeks without clinical deterioration IR2: <50% increase in SPD with new lesion(s), or ≥ 50% increase in SPD of a lesion or set of lesions at any time during treatment IR3: increase in 18F-FDG-uptake without increase in lesion size meeting criteria for PD
** RECIL (2017) **	CT: complete disappearance of all TL and all nodes with LD <10 mm PET: normalized 18F-FDG-uptake (DS 1–3)	* Partial response * CT: ≥ 30% decrease in SLD of TL, but no CR PET: DS 4 or 5 * Minor response * Same as PR yet only ≥ 10% and <30% SLD decrease	CT: <10% decrease or ≤ 20% increase SLD of TL PET: any DS	CT: >20% increase in SLD of TL For small lymph nodes <15 mm after therapy, a minimum absolute increase of 5 mm and the LD >15 mm New lesion(s) PET: any DS

CT: Computed Tomography, DS: Deauville score, ^18^F-FDG: Fluorodeoxyglucose, IR: Indeterminate Response, LD: Long diameter, PD: Progressive Disease, PET: Positron Emission Tomography, PR: Partial Response, SLD: Sum of Longest Diameters, SPD: Sum of Perpendicular Diameters, TL: Target Lesions.

**Table 3 ijms-22-04159-t003:** Response evaluation criteria in multiple myeloma.

**Lesion Type**	**Site**	**Number of Lesions (x)**	**Grading**
**Diffuse**	Bone marrow		Deauville scale (five points)
**Focal (F)**	Skull (S) Spine (SP) Extraspinal (ExP)	x = 1 (no lesions) x = 2 (1 to 3 lesions) x = 3 (4 to 10 lesions) x = 4 (>10 lesions)	Deauville scale (five points)
**Lytic (L)**		x = 1 (no lesions) x = 2 (1 to 3 lesions) x = 3 (4 to 10 lesions) x = 4 (>10 lesions)	
**Fracture (Fr)**	At least one		
**Paramedullary (PM)**	At least one		
**Extramedullary (EM)**	At least one	N/EN (Nodal/ExtraNodal) *	Deauville scale (five points)

* For nodal disease (N): C: Cervical, SC: Supraclavicular, M: Mediastinal, Ax: Axillary, Rp: Retroperitoneal, Oth: Other, Mes: Mesenteric, In: Inguinal. For extranodal disease (EN): Li: Liver, Mus: Muscle, Spl: Spleen, Sk: Skin.

## Data Availability

Not applicable.

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
