# Peer review of "PET Molecular Imaging: A Holistic Review of Current Practice and Emerging Perspectives for Diagnosis, Therapeutic Evaluation and Prognosis in Clinical Oncology"

_ijms, 2021, doi:10.3390/ijms22084159_

Round 1

Reviewer 1 Report

The Review entitled “PET molecular imaging: a holistic review of current practice and emerging perspectives for diagnosis, therapeutic evaluation and prognosis in clinical oncology” by Valentin Duclos and colleagues presents the historical evolution of PET imaging, reporting specifically on the applications of 18F-FDG PET in clinical practice. The subject of the manuscript is interesting and the manuscript is well-written and organized. Personally, I think that the manuscript is suitable for publication and that IJMS would be an appropriate place for it to be published.
I would like to suggest that the authors address the following points to improve the manuscript:

  • The use of the terms ‘radiolabeling’ or ‘radiolabeled’ is often not applied in a correct way throughout the manuscript. For example, on line 22 it is mentioned “Radiolabeling of tumor immunity”. However, radiolabeling refers to the process or technique used to incorporate a radioisotope in a ligand that binds a specific target in a relevant biological process. I recommend that these terms are replaced appropriately.
  • All the tables are missing a title. These should be added to improve readability.
  • Some acronyms should be extended for the first time: line 246 HL, NHL, DLBCL; line 292 CRAB
  • On page 15 multiple therapeutic radionuclides are mentioned. However, it would be useful to the reader to distinguish between radioligand therapy using either alpha or beta- particulate radiation. What is the general trend in the field? What are the limitations? Answering these questions will allow the reader to understand that these different types of radionuclides have different properties and consequently different specific applications.
  • On line 375 the word “relevant” should be replaced by ‘targeted’ or ‘specific’.
  • On line 423 the nomenclature of [18F]FDG should be corrected.
  • Lines 456, 457: Can you provide a context for combined the use of FDG and FAPi PET? Please present literature to support this claim.

Author Response

The use of the terms ‘radiolabeling’ or ‘radiolabeled’ is often not applied in a correct way throughout the manuscript. For example, on line 22 it is mentioned “Radiolabeling of tumor immunity”. However, radiolabeling refers to the process or technique used to incorporate a radioisotope in a ligand that binds a specific target in a relevant biological process. I recommend that these terms are replaced appropriately.
=> We thank the reviewer for this comment. We replaced these terms appropriately when necessary throughout the revised version of the manuscript.
All the tables are missing a title. These should be added to improve readability.
=> We added a title for each of the tables in the revised version of the manuscript.
Some acronyms should be extended for the first time: line 246 HL, NHL, DLBCL; line 292 CRAB
=> We extended the acronyms in the revised version of the manuscript.
On page 15 multiple therapeutic radionuclides are mentioned. However, it would be useful to the reader to distinguish between radioligand therapy using either alpha or beta- particulate radiation. What is the general trend in the field? What are the limitations? Answering these questions will allow the reader to understand that these different types of radionuclides have different properties and consequently different specific applications.
=> We thank the reviewer for this relevant comment. We clarified this important point in the legend of Figure 6 in the revised version of the manuscript.
On line 375 the word “relevant” should be replaced by ‘targeted’ or ‘specific’.
=> We replaced the word “relevant” by ‘specific’ in the revised version of the manuscript.
On line 423 the nomenclature of [18F]FDG should be corrected.
=> We corrected the nomenclature of [18F]FDG in the revised version of the manuscript.
Lines 456, 457: Can you provide a context for combined the use of FDG and FAPi PET? Please present literature to support this claim.
=> We clarified this point accordingly in the revised version of the manuscript.

Reviewer 2 Report

The review covers the current applications and the emerging perspectives of PET/CT in oncology, following a holistic approach. This is an important topic and will be of interest to a wide audience. The paper is well written and the manuscript can be recommended for publication in the present form.

Author Response

We thank the reviewer for his very positive comment.

Reviewer 3 Report

The authors have produced a very comprehensive review concerning the use of PET in modern oncology starting from technical aspect of quantitative imaging to innovative applications of new agents for theragnostic purposes.

This holistic approach is challenging because requires to condensate in few pages the overall scientific production focused on PET in the last 20 years.

The work appears complete in some sections, especially when addressed technical issues, but it results to be implemented in radiopharmaceutical aspects and clinical setting of solid tumours.

I think that a focusing on haematological malignancy excluding the other tumours may contribute to simplify the paper giving higher attraction for the readers.

In specific tables 1, 2 and 3 should be reformatted.

In the section 2.2.: The use of PET as predictor of tumour response during treatment has been initially demonstrated in gastric and rectal cancer. These references should be included

Author Response

I think that a focusing on haematological malignancy excluding the other tumours may contribute to simplify the paper giving higher attraction for the readers. In specific tables 1, 2 and 3 should be reformatted.
=> We thank the reviewer for this comment. Beyond the technical aspects developed particularly in the sections N°2 (quantification capabilities : what can we generate in term of imaging data) and N°5 (image post-processing in the era of computer science : how to deal with multidimensional data for optimized decision-making in the next few years), two others key concepts of PET imaging are developed in sections N°3 and N°4 : interpretation criteria in current practice (section N°3) and radionuclide labeling of targeted molecular probes (section N°4). To illustrate these two major concepts of PET imaging, we focused our work on current or emerging clinical practices.
  • In section N°3 (“PET molecular imaging: response evaluation criteria in practice”), we explained 20 years of history of standardization in routine PET, from the initial motivations to the current guidelines in practice, which are currently based either on visual OR semi quantitative PET metrics, depending on the targeted malignancy. For this reason, and motivated by the growing role of PET imaging to manage these diseases, we illustrated these evolving response criteria with the diversity of malignant entities they refer to: Solid tumors, Lymphoma, and more recently Multiple Myeloma, for which PET is becoming a tool of choice.
  • In section N°4 (“PET molecular imaging: the promising clinical perspective of radioligand molecular imaging and therapy”), we emphasized the vectorized PET imaging capabilities with major breakthroughs in this field: TNE and PCa. We finished the section with the promising perspective of imaging the immunity of tumors, a major field of research in the next few years.
Our goal was to give to the reader the keys to understand the general foundations of the PET imaging capabilities to adapt to the current and emerging challenges in Oncology. Consequently, focusing on a particular malignancy is not beyond the scope of this work.
We reformatted tables 1, 2 and 3 in the revised version of the manuscript.
In the section 2.2.: The use of PET as predictor of tumour response during treatment has been initially demonstrated in gastric and rectal cancer. These references should be included
=> We thank the reviewer for his comment and added 2 references accordingly in the revised version of the manuscript.

Round 2

Reviewer 3 Report

The author's prospective is better explained now.

However, the main topic in section 4 is the analysis of theragnostic applications (with exclusive attention on NET and PC) more than the promising clinical perspective of radioligand molecular imaging. The title of section 4 should be modified.

Concerning to the reply "Our goal was to give to the reader the keys to understand the general foundations of the PET imaging capabilities to adapt to the current and emerging challenges in Oncology", i think that to well understand capabilities of nuclear medicine in oncology more words should be spent on the kinetic and distribution of radiopharma. The correlation between FDG uptake and gene profile in breast cancer, may be an emerging challenge in example. The risk to appear tautological more than holistic is substantial.  However the section on the response assesment by using PET is well focused and justify the acceptance for pubblication.

Author Response

The main topic in section 4 is the analysis of theragnostic applications (with exclusive attention on NET and PC) more than the promising clinical perspective of radioligand molecular imaging. The title of section 4 should be modified.
=> We thank the reviewer for his comment. In fact, the section 4 is not “stricto sensu” focused on Theragnostic applications with exclusive attention on NET and PCa, but more largely on the general capabilities of radioligand molecular probe targeting to support oncological imaging or therapy purposes. To illustrate this key property, 3 sub-sections are thus developed because of their high relevance as breakthroughs or emerging practices in oncology: 1) NETs and 2) PCa for which PET imaging, due to its high resolution/quantification/whole body mapping capabilities, has boosted recently the image-driven radiolabeled vectorized therapy in these clinical fields; and 3) PET imaging of tumor immunity and its surrounding stroma, a very promising perspective.
Consequently, this section is not only focused on NETs and PCa, reason why we would prefer, if possible, to maintain the current title “PET molecular imaging: the promising clinical perspective of radioligand molecular imaging and therapy” which better illustrates the content of the 3 sub-sections we developed.
